# Myokines: Crosstalk and Consequences on Liver Physiopathology

**DOI:** 10.3390/nu15071729

**Published:** 2023-03-31

**Authors:** Aurore Dumond Bourie, Jean-Baptiste Potier, Michel Pinget, Karim Bouzakri

**Affiliations:** 1European Center for the Study of Diabetes (CeeD), Research Unit of Strasbourg University “Diabetes and Therapeutics”, UR7294, 67200 Strasbourg, France; 2ILONOV, 67200 Strasbourg, France

**Keywords:** NASH, NAFLD, HCC, diabetes, myokines, inter-organ crosstalk, metabolic disease

## Abstract

Non-alcoholic fatty liver disease (NAFLD) is a chronic liver disease mainly characterized by the hepatic accumulation of lipid inducing a deregulation of β-oxidation. Its advanced form is non-alcoholic steatohepatitis (NASH), which, in addition to lipid accumulation, induces hepatocellular damage, oxidative stress and fibrosis that can progress to cirrhosis and to its final stage: hepatocellular carcinoma (HCC). To date, no specific therapeutic treatment exists. The implications of organ crosstalk have been highlighted in many metabolic disorders, such as diabetes, metabolic-associated liver diseases and obesity. Skeletal muscle, in addition to its role as a reservoir and consumer of energy and carbohydrate metabolism, is involved in this inter-organs’ communication through different secreted products: myokines, exosomes and enzymes, for example. Interestingly, resistance exercise has been shown to have a beneficial impact on different metabolic pathways, such as lipid oxidation in different organs through their secreted products. In this review, we will mainly focus on myokines and their effects on non-alcoholic fatty liver disease, and their complication: non-alcoholic steatohepatitis and HCC.

## 1. Introduction

Non-alcoholic fatty liver disease (NAFLD) is a metabolic disease characterized by an over accumulation of lipids in the liver [1], also called hepatic steatosis, which can lead to several downstream complications. Affecting up to 25% of adults in the world, NAFLD has recently become a public health concern due to the increasing rates of obesity, diabetes and metabolic syndrome [2]. It is usually a silent disease, with few or no specific symptoms, and is often detected in routine tests [3]. Due to both genetic and environmental factors, the exact mechanisms underlying the pathophysiology of hepatic steatosis and its complications are not well defined. Among genetics factors, the occurrence of mutations in specifics genes such as *Pnpla3*, *Tm6sf2* and *Mboat7* have recently been described as new crucial risk factors in lipid accumulation and NAFLD onset [4]. The increase in lipid accumulation is not harmful for the liver, however, the steatosis-induced metabolic perturbations occurring within hepatocytes often lead to several other dysfunctions, such as insulin resistance, oxidative stress and even lobular inflammation [5,6].

Under these conditions, and for approximately 25% of patients, NAFLD can progress towards a more severe form—non-alcoholic steatohepatitis (NASH) [7]. Researchers have tried to explain the occurrence of the disease using the “two-hits” hypotheses, in which lipid accumulation and insulin resistance are defined as the “first hit”, and inflammation, fibrosis and oxidative stress, that we will develop as the “second hit” [8].

In this context, the hepatic microenvironment is strongly disturbed due to hepatocyte dysfunction. As results, the secretion of pro-inflammatory and pro-fibrotic factors along with the circulation of apoptotic bodies and danger signals, such as damage-associated molecular patterns (DAMPs) in the liver, will trigger other resident cell type such as Küpffer cells and hepatic stellate cells [9].

Küpffer cells are the resident liver macrophages and are differentiated from circulating monocyte. Their role is crucial in the maintenance of the innate immune response in the liver, where they can phagocytise circulating debris and particles, and help fight infection through the secretion of pro-inflammatory cytokines [10]. However, in the context of NASH, the over-activation of these cells will lead to even more lobular inflammation and will trigger other liver cell types, such as hepatic stellate cells [11]. 

Hepatic stellate cells are perisinusoidal cells that are usually quiescent in healthy individuals, and primarily store vitamin A and retinol inside of lipid droplets. In the context of Küpffer cell activation and proinflammatory cytokines circulation, these cells will transdifferentiate into activated myofibroblasts and secrete significant amounts of fibrotic factors, such as TGF-β1 or collagen, thus leading to liver fibrosis [12]. Liver fibrosis is a result of the evolution of NASH, and is defined by an accumulation of scarring within the liver, replacing healthy tissue and inducing a loss of liver function [7]. The final stage of fibrosis, cirrhosis, is associated with marked symptoms and could lead to acute liver failure and drastically increase the need for a transplantation [13,14]. To date, no pharmacological treatment is authorized for the prevention or treatment of NAFLD or NASH. The only measures recommended for reversing its progression are dietary improvement and calorie restriction, along with the practice of physical exercise [15].

Furthermore, NASH-related cirrhosis can, in a minority of patients, lead to the development of hepatocellular carcinoma (HCC), a primary form of liver cancer and the most recurrent one, representing 90% of all liver cancers. Several chronic liver diseases can lead to the development of HCC, such as alcohol, hepatitis C and D viruses and NASH. However, NASH has recently become the most common cause of HCC, due to the pandemic of obesity and diabetes.

It is well known that physical exercise has a significant beneficial impact in the management of metabolic diseases and their complications, such as obesity, metabolic syndrome, cardiovascular diseases and NAFLD [16,17]. The mechanisms by which physical activity exercises its effects are well defined, and recent topics have recently come to enrich these theories [18]. For a long time, resistance exercise was mainly known for its positive effects in maintaining and gaining muscle mass and endurance exercise was particularly encouraged due to its beneficial effects on cardiovascular health and functional capacity [19]. However, the role of myokines has recently emerged. Myokines are specific cytokines secreted by skeletal muscle cells in response to muscle contraction, notably during resistance exercise [19], and subsequently released in the bloodstream. These cytokines have been described as new crucial components of inter-organ crosstalk and whole-body homeostasis, especially concerning metabolic diseases [20,21], impacting physiological processes include insulin sensitivity [22], aging [23], glucose and lipid metabolism [24] and pancreatic beta cell function [25,26]. They can act in an autocrine, endocrine and paracrine manner on different organs such as the brain, bones, adipose tissue, pancreas or liver [27]. Only a few proteins have been described as myokines, among which the most studied ones are IL-6 [28], myostatin [29] and irisin. Interestingly, it has been described that the profile of secretion of myokines depends on the state of the individual’s health and could therefore be more deleterious among patients suffering from insulin resistance or diabetes. This suggests that the overall impact of physical activity in single individuals could be partially explained by a complex balance of beneficial and deleterious myokines, depending on the disease and state of the patient.

As already stated, liver is one of the many organs on which myokines acts. In this context, we decided through this review to gather from the literature some key proteins already described as myokines, and their documented effects on the pathophysiological aspects of NAFLD and its complications, such as NASH and HCC. 

## 2. Involvement of Myokines in NAFLD Pathology

Non-alcoholic fatty liver disease (NAFLD), or hepatic steatosis, is defined as the first hit of non-alcoholic liver steatohepatitis (NASH) and is characterized by lipid accumulation in the hepatocytes. This first hit of the disease is reversible and a relatively harmless affliction. However, lipid accumulation in the cells disturbs different biochemical processes, such as fatty acid β-oxidation increasing the secretion of pro-inflammatory and pro-fibrotic factors, which induce the progression of NAFLD to NASH with the development of inflammation through the activation of Küpffer cells (liver resident macrophages) and fibrosis in hepatic stellate cells (HSC).

Thus, preventing NAFLD is of particular interest to prevent and reduce the incidence of NASH and development of HCC.

### 2.1. Glucose Metabolism (Figure 1)

The liver is involved in glucose homeostasis by maintaining a balance between glucose production through gluconeogenesis (involving enzymes such as PCK1 and G6Pase) and glycogenolysis, as well as glucose storage through glycogenesis (via inhibition of GSK3).

Physical exercise increases peroxysome proliferator-activated receptor gamma coactivator 1-alpha (PGC1α) expression, responsible for the synthesis of fibronectin type III domain-containing protein 5 (FNDC5), and a proteolytic cleavage product of FNDC5 is known as irisin [30]. Irisin is a glycosylated type-1 membrane protein, whose expression is increased in skeletal muscle and circulation during exercise [30]. Interestingly, irisin is involved in the reduction of insulin resistance in the liver [30]. In hepatocytes, Perakakis et al. highlighted that irisin decreases gluconeogenesis by reducing *PCK1* and *G6PC* expressions through the PI3K/AKT/FOXO1 pathway [31]. In parallel, irisin also increase glycogenesis by activating the PI3K/AKT/GSK3 pathway [31]. Irisin also has an impact through the decrease of cholesterol expression by reducing SREBP2 expression in hepatocytes [31].

IL-6 is a cytokine, whose main source comes from contracting skeletal muscle and increases in circulation immediately after exercise [32]. Banzet et al. highlighted the implication of IL-6 secretion on the control of hepatic glucose production, and more precisely on gluconeogenesis by inducing an increase of PCK1 expression [33]. Furthermore, Pedersen et al. underlined that IL-6 secretion is not only dependent on exercise but is also inversely related to plasma glucose level [34], suggesting that IL-6 secretion aims to induce glycogenolyse in the liver to respond to the lack of circulating glucose, for example in the case of overnight fasting [34]. Thus, IL-6′s impact on glucose production is dependent on the metabolic state [35]. Furthermore, another muscle-liver crosstalk has also been highlighted by Pedersen et al., who showed that exercise-induced liver CXCL1 cytokine expression is dependent on physical exercise-derived skeletal muscle IL-6 secretion [34]. However, the role of CXCL1 in liver glucose metabolism is not yet fully understood. The role of IL6 was hereby mostly described as beneficial and crucial for liver homeostasis. However, this cytokine is mostly known for being secreted during the acute phase of infection, mostly by activated Küpffer cells in the liver [36]. Indeed, other studies highlighted the properties of IL-6 to impair insulin signaling and glucose homeostasis, mostly through JNK1, STAT3 and SOCS3 signaling [37,38,39]. These data demonstrate a bimodal and pleiotropic role for IL-6 in liver homeostasis, and suggest that further studies should be carried out, not only using IL-6 knock-out models but also with the use of cell-specific depletion, or with different dose of recombinant protein, to define the exact impact of IL-6 signaling in the liver, and especially its role as a myokine in this context.

Another member of the interleukine-6 cytokine family is the Leukemia Inhibitory Factor (LIF), mainly expressed in the white adipose tissue, but it has also been identified in the muscle. LIF and its receptor, LIFR, are involved in glucose metabolism through the inhibition of gluconeogenesis [40].

The Fibroblast Growth factor 21 (FGF21) is mostly known as a hepatokine [41], but it has also been found to be expressed in skeletal muscle, where it is regulated by the insulin pathway (AKT/PI3K) [42]. Furthermore, muscle-derived FGF21 exerts metabolic actions, notably on the liver [43]. Gong et al. highlights that FGF21 improves hepatic insulin sensitivity through the inhibition of the mammalian target of rapamycin complex 1 (mTORC1) [43]. mTORC1, by activating its downstream effector, the ribosomal protein S6 kinase 1 (S6K1), induces the IRS1 phosphorylation at serine 101 and 302, that are responsible for the induction of insulin resistance [43]. Furthermore, Gong et al. showed that the administration of FGF21 in vitro and in vivo increases hepatic glycogen synthesis [43].

Myostatin is a member of the TGF-β superfamily and is the first myokine identified in 1997 by Se-Jin Lee et al. [44]. It is one the rare myokines, whose expression is reduced during exercise [45]. Myostatin induces the reduction of muscle mass through the decrease of factors involved in tissue growth [45], which leads to the reduction of hepatocytes’ sensitivity to insulin via the decrease of IRS1 and 2 expression [46]. 

**Figure 1 nutrients-15-01729-f001:**
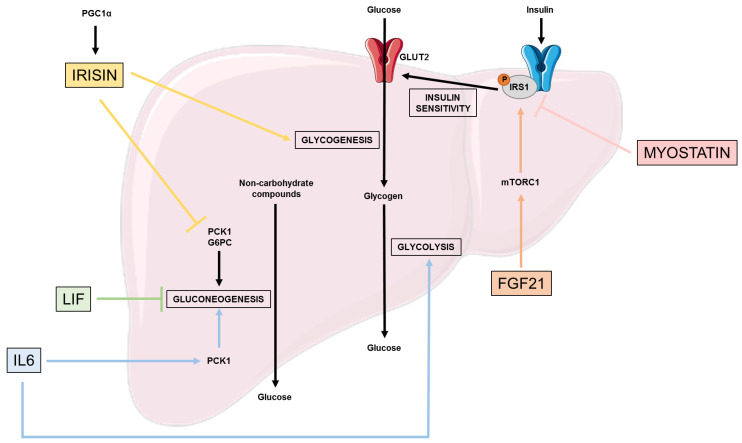
Effects of myokines on glucose metabolism. This figure represents the beneficial and deleterious effects of different myokines on the different pathways involved in glucose metabolism: insulin sensitivity, glycogenesis, glycolysis, and gluconeogenesis. FGF21: Fibroblast growth factor 21; G6PC: Glucose-6-phosphatase catalytic subunit; GLUT2: Glucose transporter 2; IRS1/2: Insulin receptor substrate 1/2; LIF: Leukemia inhibitory factor; mTORC1: Mammalian target of rapamycin complex 1; PCK1: phosphoénol pyruvate carboxykinase; PGC1α: Peroxysome proliferator-activated receptor gamma co-activator 1-alpha.

### 2.2. Lipid Metabolism (Figure 2)

Lipid metabolism is also critical for glucose homeostasis. In physiological conditions, the liver converts excess fatty acid into triglycerides, their hepatic storage form, through de-novo lipogenesis (DNL) [47]. To compensate this storage, PPARα regulates the conversion of fatty acids into energy through the mitochondria fatty acid β-oxidation, a pathway reducing hepatic fat accumulation and secreting very-low-density lipoprotein (VLDL) exporting lipids to other tissues [47]. However, in pathological conditions and excess lipids in the liver, the existing pathways to compensate this excess no longer work.

Irisin also has an impact in reducing lipogenesis, by decreasing the expression of lipogenic markers such as *Acaca* and *Fasn,* and lipid accumulation [31]. However, among patients with hepatocellular carcinoma (HCC), irisin circulating levels are associated with higher NAFLD incidence through an increase in the expression of key de-novo lipogenesis proteins such as SCD1 and SREBPF1. Further studies are needed to fully understand the role of irisin, which is either involved in over-expression or over-secretion to counteract the DNL proteins and reduce them [40].

IL-15, which is considered as one of the most frequently described myokines [48], has been described to be implied in lipid accumulation related to HFD [49]. Using a KO mouse model lacking IL-15 or its receptor IL-15Rα, Cepero et al. showed that the animals were accumulating less lipid compared to the control, leading to a prevention of hepatic steatosis.

Myonectin, also a myokine secreted in response to physical exercise, increases the expression of proteins involved in liver fatty acid uptake, such as the cluster of differentiation 36 (CD36), fatty acid transport protein (FATP) and fatty acid binding protein 1 and 4 (FABP) [21,50].

LIF expression in the circulation is correlated with the severity of liver steatosis [51]. However, this circulating LIF overexpression in HFD mice is associated to the alleviation of hepatic steatosis, notably via the inhibition of SREBP1-c expression through the activation of STAT3 pathway reducing hepatic triglycerides accumulation [40]. LIF exerts these effects through its binding onto its receptor, LIFR, expressed on different cell types such as hepatocytes, which is down-regulated in the obese state and NAFLD patients [40]. Thus, targeting the LIF-LIFR couple is of interest to improve NAFLD treatment.

### 2.3. Effects on β-Oxidation and Oxidative Stress (Figure 2)

Interestingly, irisin also has an impact on fatty acid oxidation through its main target PPARα [52], and it decreases oxidative stress by reducing the expression of inflammatory factors, such as NF-kB, p38 MAPK, TNF or IL6, but also the production of reactive oxygen species (ROS) in hepatocytes [31].

IL-6 myokine has been shown to alleviate diet-induced liver steatosis [53] through an increase in mitochondrial fatty acid β-oxidation [54,55]. More precisely, IL-6 exerts positive effects on PPARα, which increases the expression of carnitine palmitoyltransferase 1 (CPT1) involved in fatty acid oxidation [56].

Β-aminoisobutyric acid (BAIBA) is a PGC1α-dependent myokine, secreted by the muscle when PGC1α is expressed in the tissue [57]. PGC1α expression is induced by physical exercise, which activates the peroxisome proliferator-activated receptor α (PPARα). Thus, BAIBA is involved in the improvement of hepatic lipid metabolism through PPARα-induction of β-oxidation and decreases lipogenesis [57].

On the other hand, myostatin alters hepatic lipid metabolism, increasing the triglycerides levels in the hepatocytes through an increase in FAS expression and suppression of PPARα [46].

To conclude, the majority of myokines secreted by the muscle during physical exercise have beneficial effects on glucose homeostasis through the increase of hepatocytes’ insulin sensitivity, and on lipid metabolism through the decrease of triglycerides excess storage; notably through the increase of mitochondria β-oxidation, and the reduction of oxidative stress and of excessive de-novo lipogenesis.

**Figure 2 nutrients-15-01729-f002:**
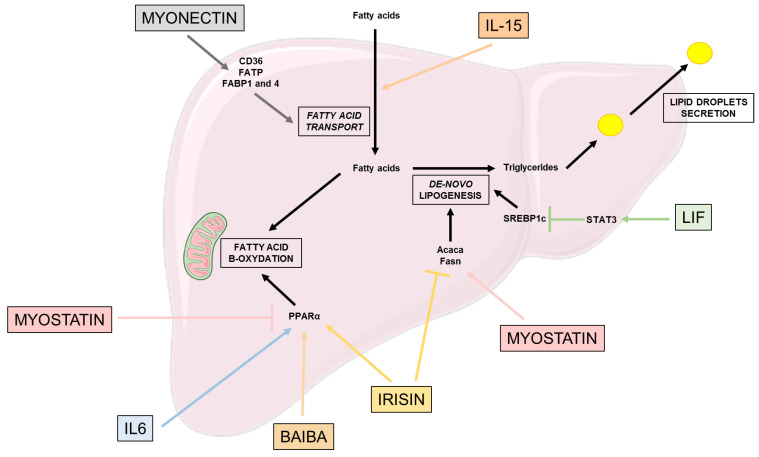
Effects of myokines on lipid metabolism. This figure highlights the beneficial and deleterious effects of different myokines on the different pathways involved in lipid metabolism: fatty acid transport, de-novo lipogenesis, fatty acid β-oxidation and lipid droplet secretion. Acaca: acétyl-CoA carboxylase 1; BAIBA: β-aminoisobutyric acid; CD36: Cluster of differentiation 36; FABP: Fatty acid binding protein; FATP: Fatty acid transport protein; Fasn: Fatty acid synthase; IL-6: Interleukine 6; IL-15: Interleukine 15; LIF: Leukemia inhibitory factor; PPARα: Peroxisome proliferator-activated receptor alpha; SREBP: sterol regulatory element binding protein; STAT3: Signal transducer and activator of transcription 3.

## 3. Involvement of Myokines in NASH Pathophysiology

NASH is the second hit of the NAFLD-NASH hallmarks and occurs when hepatic lipid accumulation and hepatocyte dysfunction induce the secretion by hepatocytes of pro-inflammatory and pro-fibrotic factors along with DAMPS. These secreted proteins have an impact on the other hepatic cell types:-Küpffer cells, the liver resident macrophages, generally in a non-inflammatory polarization are activated by pro-inflammatory factors to a M1 polarization responsible for an inflamed state in the liver;-Hepatic stellate cells (HSC), usually quiescent, are activated and transdifferentiate into myofibroblasts, which are responsible for fibrosis formation.

Thus, in this part we will discuss the role of myokines and of the main hallmarks of NASH: inflammation and fibrosis.

### 3.1. Inflammation

Decorin, a myokine part of the small leucine-rich proteoglycan family, is known for its anti-inflammatory and anti-fibrotic effects [58]. Zheng et al. studied the relevance of ursolic acid (UA), a bioactive components from a medicinal plant and an inhibitor of decorin, in the treatment of NASH [58]. This work highlights the involvement of decorin in reducing hepatic inflammation and fibrosis regulation, notably through the activation of the insulin growth factor I receptor (IGFIR) and the inhibition of hypoxia inducible factor 1 (HIF-1) [58].

Despite the positive effect of IL-15 on steatosis previously explained, the suppression of IL-15 in hepatocytes both in vivo and in vitro induced a significant decrease in inflammation, especially regarding the expression of chemokines, which is possibly related to the diminution of natural killer (NK), natural killer T cells (NKT), and the invariant natural killer T cells (iNKT) cell macrophages infiltration [49].

Polyzos et al. highlighted an independent but positive correlation between high plasma irisin levels and a severe inflammation in NAFLD patients that could be explained by a compensatory mechanism by irisin to induce an anti-inflammatory response [59].

In addition to improving hepatic insulin sensitivity, low myostatin plasma levels reduce the expression of tumor necrosis factor α (TNFα), a factor involved in the progression of NAFLD to NASH through the induction of hepatic inflammation [60,61].

### 3.2. Fibrosis

Even if IL-15 suppression downregulates liver inflammation, its long term depletion induces adverse effects such as the diminution of cell types such as natural killer and natural killer T cells, which could be an issue. Indeed, Jiao et al. showed that the knockout of IL-15Rα in mice led to an aggravation of liver fibrosis, both by directly activating HSC and reducing the anti-fibrotic effects of NK cells [62].

The effects of myostatin, a member of the TGF-β superfamily, are close to those of TGF-β, notably concerning the growth arrest of hepatic stellate cells and the overexpression of factors involved in fibrogenesis and to procollagen secretion by HSC; but remains independent of TGF-β1 expression in HSC [63]. Thus, the main implication of myostatin in fibrosis is its capacity to activate HSC into fibroblasts; however, in terms of mechanisms of action, they are still not fully understood. Myostatin’s receptor ActR2B is expressed on HSC and hepatocytes, furthermore an activation of JNK-1 is consistent with myostatin expression and is involved in fibrogenic process through the activation of downstream pathways such as cell proliferation, oxidative stress and the expression of pro-fibrotic genes [63]. Thus, myostatin-JNK-myofibroblasts activation axis is of interest for the understanding of myostatin’s role in the activation of HSC to fibroblast, and its implication in fibrosis.

Similarly to Polyzos et al. [59], Armandi’s team described an independent but positive correlation between irisin-increased expression and liver disease, with an increase in fibrogenesis [64] associated with an increase in collagen remodeling factors, such as PRO-C3 and PRO-C6. 

Apelin is a myokine, whose expression is increased by endurance training on skeletal muscle [65]. Different sizes of the *C*-terminal fragments give different forms of apelin [66]; here we focus on apelin-13. Wang et al. demonstrated that increased levels of apelin-13, in NAFLD murine models, induce the upregulation of collagen type-I, α-smooth muscle actin (α-SMA), and cyclinD1 in human hepatic stellate cells (LX-2) [67]. Furthermore this fibrogenic gene upregulation goes along with an increase in pERK1/2 expression in LX-2 cells; however, the mechanisms linking apelin-13 to the upregulation of fibrogenic genes through ERK1/2 phosphorylation are still not known [67].

To conclude, the majority of myokines involved in the control of inflammation seem to have beneficial effects (Figure 3). However, in terms of fibrosis, further studies are needed to determine if the fibrosis-related myokines high plasma levels are linked to deleterious or to compensatory effects.

## 4. Myokines Effects on Hepatocellular Carcinoma

Hepatocellular carcinoma is a primary liver cancer that can occur as an evolution of several chronic liver diseases. It is the most common form of liver cancer, accounting for approximately 90% of all liver cancer. In general, the evolution of liver diseases to cirrhosis is a leading risk factor of HCC development [68]. As previously said, NAFLD and its progression to NASH are a new emerging cause of liver cirrhosis and HCC [69]. Thus, we will hereby discuss the impact of some previously studied myokines and their potential impact or relation to HCC development.

Decorin expressed by cancer-associated fibroblast (CAF) has been shown to be important in the inhibition of HCC through a synergistic inhibition with integrin β1 of HCC cell invasion and migration [70]. Thus, Zheng et al. presents cancer-associated fibroblast-related decorin, a promising strategy for the clinical treatment of HCC [70].

IL-15′s ability to positively modulate NK cells also has an impact in the context of liver tumor development and HCC. Indeed, IL-15 was shown to restore the function of NK cells in vitro, thus ameliorating the anti-tumoral response [71]. Another study, focusing on HCC patients, derived in vitro cultures, similarly highlighted the potential of IL-15 to enhance NK cell response, suggesting its potential role as an immunotherapy targeting HCC [72]. However, to fully elucidate the role of IL-15 as a myokine, further studies are needed to determine the correspondence between exercise-stimulated IL-15 plasma levels and the concentration used in the previously cited studies.

Interleukin-6 is important for proper hepatic homeostasis and liver regeneration, but its constant over-activation could lead to HCC development through the activation of the JAK and gp130 pathway [73,74].

Myostatin serum levels are associated with severe forms of cirrhosis in humans and decreased hepatic function, such as albumin production [75]. Furthermore, among patients suffering from NAFLD-derived HCC, who underwent primary surgical resection, serum myostatin were associated with lower overall survival, and, as previously said, with the severity of liver fibrosis [76].

Concerning irisin, its gene expression was correlated with the expression of Notch1, a key driver of HCC metastasis, among HCC suffering patients [77]. However, further studies must clarify whether irisin is responsible for this effect or overproduced as a protective hormone to counter the severity of the disease [40].

To conclude, we hereby discussed the potential relationship of few myokines with HCC incidence or development (Figure 4). However, not many studies exist on the effects of myokines on NAFLD-related HCC. Thus, further investigations are required to fully elucidate the involvement of physical exercise and myokines in this context.

## 5. Conclusions

In this review, we have discussed the involvement of myokines, muscle-secreted cytokines, in non-alcoholic fatty liver disease (NAFLD) pathology and its progression to non-alcoholic steatohepatitis (NASH) and hepatocellular carcinoma (HCC). 

NAFLD is the most prevalent form of chronic liver disease and is mostly correlated with type-2 diabetes and obesity, with a prevalence of 70% and 80%, respectively [78]. Despite its high prevalence, no specific treatment exists to date. The first recommendation, as for many metabolic diseases, is lifestyle improvements—such as dietary improvement or physical exercise. Indeed, physical exercise enables the secretion of cytokines by the muscle, called myokines, which have autocrine and paracrine effects on different tissues. In this review, we highlighted the importance of these myokines in the regulation of NAFLD’s different stages. The majority of myokines have beneficial effects on the different hallmarks of NAFLD, NASH and HCC. In the context of NAFLD, many myokines have the capacity to control glucose homeostasis by activating or inhibiting glucose metabolism pathways, such as insulin sensitivity, glycogenesis, gluconeogenesis and glycolysis; thus, having an impact on both glucose storage, in the form of glycogen, and on its production from glycogen or non-carbohydrate compounds for energy needs. Those myokines and others are also enabled to impair the development of NAFLD through their capacity to control lipid metabolism pathways, such as fatty acid uptake, but also their conversion into energy through mitochondria β-oxidation, or into triglycerides through de-novo lipogenesis. For the second stage of the disease, NASH, the majority of the myokines studied in the context of inflammation present beneficial effects for its reduction. However, in the case of fibrosis, the different studies highlight an over-expression of the myokines correlated with the increase of NASH-related fibrosis. Thus, further studies are required to determine if these over-expressions are linked to the activation of pathways involved in fibrosis, or if these myokines are upregulated to counteract the mechanisms of fibrosis. Finally, in terms of hepatocellular carcinoma, not many studies are focusing of this NAFLD stage yet, and further investigations are needed to fully conclude on the role of myokines in HCC cell invasion, migration and metastasis. Though some of their mechanisms of action are still not fully understood, myokines present beneficial characteristics for the control of NAFLD, NASH and HCC, and may become interesting therapeutic treatments for these liver diseases.

Another positive aspect highlighted in this review is that the majority of myokines do not have an impact only on one hallmark of the disease but present an effect on different aspects of it. Presently, this characteristic is of high importance in the development of therapies. Indeed, today, the main cause of therapeutic failure is the specificity of treatment to one hallmark of the disease. However, most of the diseases present a wide array of characteristics, for example, with insulin resistance, lipid accumulation, inflammation, fibrosis, and HCC cell migration, invasion and metastasis. Thus, today’s challenges in therapy development are to find one treatment able to act on different disease’s aspects; first, to limit those treatment failures, and second, to as much as possible avoid side effects that can be increase with the use of two or more treatments at the same time. Thus, myokines, such as irisin or myostatin (non-exhaustive list), could respond to those therapy development criteria for the treatment of NAFLD and its progression.

Finally, in this review we focused on the cytokines secreted by the muscle and having an effect on the liver, however, physical exercise also has an impact on the over-expression of liver-secreted cytokines, the hepatokines, which in turn, can have beneficial autocrine effects on the liver. For example, hepatic follistatin, whose expression is increased in NAFLD patients, inhibits lipid accumulation and lipogenesis through the mTOR pathway [79]. Thus, according to the effects of physical exercise on the expression of liver cytokines and of other organ-derived cytokines, it could be of interest to also study their effects on NAFLD’s different stages, to explore the range of possible NAFLD therapeutic strategies. 

To conclude, although further studies are needed to fully understand their role in the different stages of NAFLD, myokines represent an interesting target for the development of new NAFLD therapeutics. In addition, physical exercise also seems to promote cytokine secretion by other tissues, such as adipose tissue, or as previously mentioned, the liver. These other different factors, called adipokines and hepatokines, may also be of interest in the management of NAFLD and participate in the positive effect of inter-organ crosstalk in the context of metabolic diseases.

## Figures and Tables

**Figure 3 nutrients-15-01729-f003:**
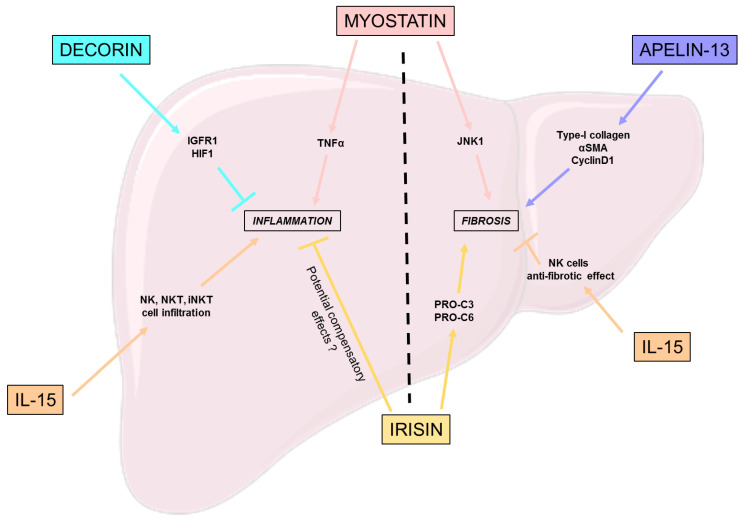
Effects of myokines on NASH hallmarks. This figure represents the beneficial, but also the deleterious effects of different myokines on the different hallmarks of NASH: inflammation and fibrosis. HIF-1: Hypoxia inducible factor 1; IGFRIR: Insulin growth factor I receptor; IL-15: Interleukine 15; JNK-1: c-Jun *N*-terminal kinase 1; NK: Natural killer; α-SMA: Alpha-smooth muscle actin; TNF: Tumor necrosis factor.

**Figure 4 nutrients-15-01729-f004:**
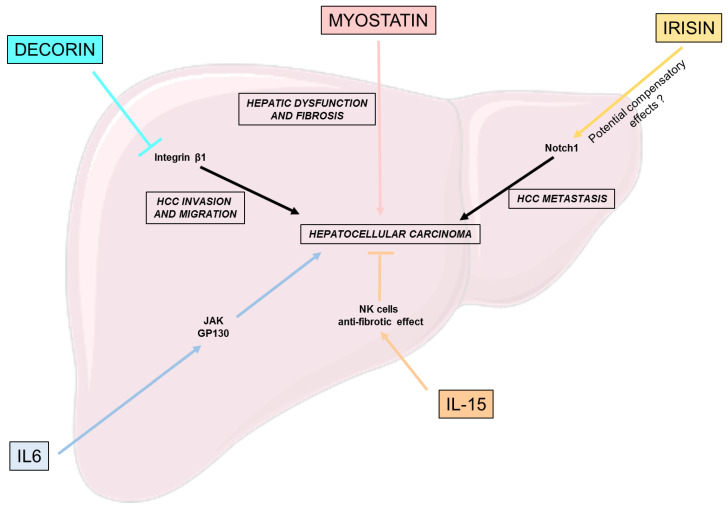
Effects of myokines on hepatocellular carcinoma. This figure puts a light on the beneficial and deleterious effects of different myokines on the different pathways involved in the development of hepatocellular carcinoma: hepatic dysfunction, fibrosis, HCC cell invasion, migration and metastasis. gp130: Glycoprotein 130; HCC: Hepatocellular carcinoma; IL-6: Interleukine 6; IL-15: Interleukine 15; NK: Natural killer.

## Data Availability

Not applicable.

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
