# Peer review of "Myokines: Crosstalk and Consequences on Liver Physiopathology"

_nutrients, 2023, doi:10.3390/nu15071729_

Round 1

Reviewer 1 Report

The study by Dumond Bourie and colleagues aims to review the role of myokines on liver physiopathology.

In general, the paper is well designed, organized and presented.

However, some aspects of the topic should be included into the manuscript and some other better discussed.

In this respect, IL-15, a myokine involved in inflammatory response and non-alcoholic fatty liver disease and showing dualistic effects, must be included and discussed into the paper.

Further, the secretion of each myokine, and its relative ratio, is influenced by the mode of training (ref. PMID: 35408868). This aspect should be briefly included and discussed as well.

In addition, the Authors present IL-6 as a beneficial cytokine for liver physiology. However, dysregulated continual synthesis of IL-6 plays a pathological effect on chronic inflammation and autoimmunity. It should be better clarified how muscle-derived IL-6 can act as a protective factor without exerting detrimental effects.

Author Response

First of all, we would like to thank you for your very interesting comments and suggestions to improve our review on the effects of myokines on NAFLD pathology.

All changes have been highlighted in yellow in the text. Please see the attachment.

Line 99 to 104: Precisions on the different type of exercises, endurance versus resistance, and on the secretion of myokines by the muscles have been added.

Concerning Il-15, its effects on NAFLD (line 221 to 225), on NASH (line 296 to 300 and line 310 to 314) and on HCC (line 365 to 372) have been added and its role has also been added to the figures 2, 3 and 4.

Line 167 to 176: The role of IL-6 in impairing liver glucose homeostasis has been described.

We hope that these added information would answer to your comments.

Regards,

Dr Dumond Bourie

Reviewer 2 Report

This is an extensive review and biochemically oriented review on the effect of myokines in the liver. In a biochemical perspective and maybe on the physiology point of view it is correct. However, major flaws are evident from the point of pathology perspective.

I would advise changing the tittle, because only few inferences on pathology are made and the few are incorrect. The review should be overall reduced and more straight to the point.

Major points:

Abstract. Remove the “which only issue is transplantation. This is not discussed in the review and it is incorrect. Due to the shortage of organs for transplant, HCC patients are often on the back of the list. For a review see World J Gastroenterol. 2019 Jun 7; 25(21): 2591–2602.

Abstract should not include terms not used in the manuscript such as MAFLD. There is no point in using acronyms if they are not used to replace the words.

I would recommend introducing a list of abbreviatures.

Line 109: liver macrophages are called Kupffer cells, or at most stellate sinusoidal macrophages, if the authors want to avoid eponyms. Liver macrophages are not named as such for centuries! This is a major error in basic Histology.

The paragraph from 103 to 116 should be reduced.

The acronyms are missing in figure legends

Author Response

Firstly, we would like to thank you for your very interesting remarks to improve our review on the myokines and the effect of diabetes-related NAFLD pathology.

All changes have been highlighted in yellow in the text. Please see the attachment.

Line 8: The term MAFLD in the abstract has been replaced by NAFLD.

Line 12: The sentence “which only solution is transplantation” has also been removed.

Line 23: List of abbreviations has been added.

Line 130: “liver macrophages” has been replaced by “Küpffer cells (liver resident macrophages)”

Abbreviations have been added at the beginning of the review and for the different figures.

Introduction of paragraph 2 has been reduced.

We hope that these added information would answer to your comments.

Regards,

Dr Dumond Bourie

Reviewer 3 Report

Please see my comments below-

- Would gender and age have an effect on expression pattern of myokines ?

- Are there any known medications known to interfere with expression of myokines.

Author Response

We would like to thank you to have accept to review our work on the impact of myokines on non-alcoholic fatty liver diseases and for your comments. Please see the attachment.

- Would gender and age have an effect on expression pattern of myokines ?

No global study on the influence of gender exists, but some studies focus on the expression of one myokine according the gender. For example, Lagzdina et al. highlight that the level of irisin concentration is associated on body-fat and lean tissue amount and dependent on gender differences (PMID: 32512797). In the study of Jia et al., IL-6, PGC1α, PPARα and myostatin expression seem to be influenced by a gender-related manner (PMID: 31056256).

- Are there any known medications known to interfere with expression of myokines.

To date, no existing treatment are specifically described to be used to impact the secretion of myokines. However, according to our review, treatments targeting myokines should be of great interest in view of their impact on non-alcoholic fatty liver diseases.

Enclosed, you will find our manuscript with some modifications highlighted in yellow following the different review reports.

Regards,

Dr Dumond Bourie
